# Hydrophilic and Functionalized Nanographene Oxide Incorporated Faster Dissolving Megestrol Acetate

**DOI:** 10.3390/molecules26071972

**Published:** 2021-03-31

**Authors:** Mohammad Saiful Islam, Faradae Renner, Kimberly Foster, Martin S. Oderinde, Kevin Stefanski, Somenath Mitra

**Affiliations:** 1Department of Chemistry and Environmental Science, New Jersey Institute of Technology, Newark, NJ 07102, USA; mi238@njit.edu (M.S.I.); fr58@njit.edu (F.R.); 2Bristol Myers Squibb Research and Early Development, Princeton, NJ 08543, USA; kimberly.foster@bms.com (K.F.); martin.oderinde@bms.com (M.S.O.); kevin.stefanski@bms.com (K.S.)

**Keywords:** enhanced dissolution, nano graphene oxide, oral medication, paddle method, gastrointestinal pH, megestrol acetate

## Abstract

The aim of this work is to present an approach to enhance the dissolution of progestin medication, megestrol acetate (also known as MEGACE), for improving the dissolution rate and kinetic solubility by incorporating nano graphene oxide (nGO). An antisolvent precipitation process was investigated for nGO-drug composite preparation, where prepared composites showed crystalline properties that were similar to the pure drug but enhanced aqueous dispersibility and colloidal stability. To validate the efficient release profile of composite, in vitro dissolution testing was carried out using United States Pharmacopeia, USP-42 paddle method, with gastric pH (1.4) and intestinal pH (6.5) solutions to mimic in vivo conditions. Pure MA is practically insoluble (2 µg/mL at 37 °C). With the incorporation of nGO, it was possible to dissolve nearly 100% in the assay. With the incorporation of 1.0% of nGO, the time required to dissolve 50% and 80% of drug, namely T_50_ and T_80_, decreased from 138.0 min to 27.0 min, and the drug did not dissolve for 97.0 min in gastric media, respectively. Additionally, studies done in intestinal media have revealed T_50_ did not dissolve for 92.0 min. This work shows promise in incorporating functionalized nanoparticles into the crystal lattice of poorly soluble drugs to improve dissolution rate.

## 1. Introduction

Liquid and solid dosages are the most convenient and widely accepted forms of oral administration in the pharmaceutical industry. However, new chemical entities (NCEs) intended for oral drug delivery are often hydrophobic and practically insoluble in water. Their therapeutic effects are significantly reduced due to low aqueous solubility, membrane permeability, and chemical and enzymatic stability [1]. These poorly water-soluble drugs are unable to fully release in the gastrointestinal tract, thus contributing to their low and variable bioavailability. NCEs with minimal solubility belong to the Biopharmaceutical Classification System (BCS) II and IV families. It is reported that over 70% of discovered drugs and active entities are categorized in these two classes. These drugs are poorly soluble, with reduced dissolution rates that suffer from formulation challenges [2]. Therefore, it is worth exploring new methods to optimize dosage forms for improved aqueous dissolution which may increase drug absorption.

Different top-down and bottom-up approaches have been explored to improve aqueous solubility for drug formulation, including the dispersion technique [3], micronization [4], nano-crystallization [5], salification [6], cyclodextrin inclusion [7], co-crystallization [8], micelle solubilization [9], solid dispersion [10], the liquisolid technique [11], nanoparticle encapsulation [12], and antisolvent precipitation [13,14]. Some of these current methods have limitations. For example, the materials cost may be high, the method can be time-consuming, or the method can alter drug morphology.

The development of successful drug formulations based on nanoparticles has opened the door for addressing the treatment of different incurable and challenging diseases. The use of nanotechnology, particle size manipulation, surface modification, and material selection have accelerated the development of nanoparticle-based immunotherapy and drug delivery [15,16,17,18]. Different functional nanostructures, including liposomes, polymers, dendrimers, carbon materials, silicon, and magnetic nanoparticles, have been verified as cell-specific carriers in drug delivery systems [19]. One such example is dendronized systems (e.g., polymers, nanoparticles, liposomes) as a way to improve the therapeutic efficiency of drugs for the treatment of cancer [20].

Two-dimensional (2D) graphene oxide (GO), with its unique chemical and mechanical properties, is emerging as a promising biomaterial for drug delivery and other biomedical applications [21]. Its excellent physiological stability, solubility, and drug loading capacity result from the unique structural properties of GO, such as sp^2^ hybridization and high surface area (2630 m^2^ g^−1^) [21]. The presence of active COOH and OH groups onto the basal and planer structure of GO produce strong conjugations with drugs through physical and chemical adsorption [22]. Therefore, GO has been used as a nanocarrier for a variety of therapeutics, anticancer medications [23], and antimicrobials [24], as well as antibodies [25], peptides, DNA, RNA, and gene delivery drugs [26,27,28,29,30]. Whereas questions have been raised about the toxicity of GO, several studies found it to be safe, with low cytotoxicity [31,32,33] and suitable for medical applications [34,35,36,37,38]. A few papers have reported the development of GO composites for drug delivery, which have found GO to be nontoxic [35,39,40].

Megestrol acetate (MA) 17-(acetyloxy)-6-methyl-pregna-4,6-diene-3,20-dione is a commercially available pharmaceutical product. It is a synthetic progesterone oral suspension which is intended for AIDS, anorexia, and cancer patients as treatment for gaining in body weight and appetite improvement [41]. MA is also considered to be an antineoplastic agent that is widely used for breast, endometrial, and prostate cancer treatment [42,43,44]. According to the BCS-II system, MA has very low solubility in water (<2.0 µg/mL), with high permeability [45]. Different pharmaceutical approaches have been used to improve solubility, including nanocrystal formulation [42], supercritical antisolvent process [46], microparticles, nano emulsions, nanosuspensions, and solid dispersions formulation [47,48,49,50,51]. The formulation techniques cited above have found variations in structural morphology from their original drugs due to the surfactant use. Altered crystal habits have also been observed. In dissolution studies, some formulations have shown reduced rates early on due to structural alterations.

The objective of this work was to synthesize MA-GO drug composites in an effort to increase its dissolution performance. MA is practically insoluble in water. Previous approaches have either altered the structure of MA or reduced its dissolution rate. Thus, a method that can effectively improve dissolution and maintain certain properties would be of interest. Our proposed method is a relatively simple antisolvent technique to improve and prepare poorly soluble MA, leading to enhanced dissolution rate.

## 2. Results and Discussion

### 2.1. Nanographene Oxide Study

The particle size of GO had to be in the nano scale for successful incorporation and enhanced solubility. Here, we produced a uniform colloidal solution of nanosized GO (or nGO) from a supplied GO with micron sized sheets. Solution was sonicated for 1.0 h using a high-power probe sonicator, maintaining the operating condition at 70% power out of 800 watts to produce 100–200 nm-sized sheets particles. The hydrodynamic Z average intensity weighted nGO particle size was confirmed to be an average of 131.4 nm (Figure 1a) on Zetasizer (nGO water solutions 1, 2 and 3 refer to three different nGO dilutions to prepare drug samples). Additionally, Atomic Force Microscopy (AFM) measurements have shown nGO particle thickness to be 1–3 nm [52]. The dynamic light scattering percent intensity distribution for MA and MA-nGO particle analysis data suggest that before (pure MA) and after antisolvent coprecipitation (MA-nGO composites) produce uniform drug composite crystallites. The Z average was found to be 2040 ± 25 (SD), 2144 ± 5 (SD), and 2157 ± 11 (SD) nm in diameter at Polydispersity Index (PDI) of 0.559 to 0.731 (Figure 1b).

Next, nGO elemental composition was studied using SEM-EDS analysis (Figure 2a). It was found that the oxygen content was almost 50.0% (Table 1). This contributed to high hydrophilicity and aided in strong intermolecular interaction with drug molecules.

### 2.2. Drug Composites Surface Morphology Analysis

The scanning electron microscope (SEM) for pure MA serves as a reference to compare with SEM of MA-nGO composites in Figure 2. It is clear from the scanning images that the crystal structure remained intact and that nGO incorporation was successful. Figure 2c–e shows the morphology of nGO on the surface of the MA drug, where the nGO sheets are spread out and produce hydrophilic linkages through H-bonding interaction. The nanosheets can also be seen attached and loosely embedded. This kind of surface phenomena can be described as activating the water channeling into the drug crystal’s surface, which contributes to higher water dispersion. Unaltered crystalline structure will maintain active ingredients polymorphism properties. Micron-sized graphene oxide was also incorporated and was found to wrap the crystal, making the composite insoluble [52].

In Figure 3, TEM image analysis was done on MA-nGO-1.09. The image shows the inner structure of the drug incorporated with nGO.

### 2.3. nGO-Drug Composite

nGO is a hydrophilic nanomaterial and forms a stable dispersion in various solvents. It shows strong hydrogen bonding at the solvent interface [53]. Many functional modifications of nGO are possible for functional interaction and complex formation in drug delivery and other biological applications. Here, we used nGO without any functional modifications, which was incorporated into drug crystals.

MA is highly hydrophobic (BCS-II drug) and insoluble in aqueous mediums. The incorporation of nGO increased the water solubility of MA-nGO to 6.6 mg/L in 24.0 h. Figure 4a (upper) shows pure drug and composites dissolved in water. Pure MA did not dissolve, whereas the composites did. It is evident from the data that as the nGO incorporation increased, the solubility also increased. This was because the nGO dispersed around the drug crystal channeled the water via hydrogen bonding.

A 1-octanol/water partitioning study (Figure 4a, lower) was done to provide further insight on physicochemical properties of formulated composites which could affect bioactivity such as absorption, distribution, metabolism, and excretion [54]. The logP value obtained from this experiment, also known as the partitioning coefficient, is a measure of hydrophobicity. It is known to correlate with oral drug solubility and interact with the phospholipid membrane. The calculated logP values for the MA-nGO composites were found to be 1.17, 1.25, and 1.36 for MA-nGO-1.04, MA-nGO-1.05, and MA-nGO-1.09, respectively, compared to 3.27 for pure MA. This suggested that the MA composites were less hydrophobic than the pure drug. Hence, their ability to dissolve in the aqueous layer was reduced.

Zeta potential measurements by Zetasizer instrumentation were used to measure surface charge of MA-nGO composites. The observed data presented in Figure 5 shows an increase in negative zeta potential values for all the MA-nGO composites, which demonstrates its higher colloidal stability compared to pure MA. The MA-nGO with lower nGO incorporation (~1.05% concentration) showed the highest stability −39.9 mV due to the uniform distribution of nGO.

### 2.4. Properties of MA-nGO Composites

Figure 6 shows the Raman spectrum of pure MA and MA-nGO composites with various nGO incorporated concentrations. The D-band region at 1340 cm^−1^ and G band at 1580 cm^−1^ from graphene oxide overlapped with peaks from MA drug. The spectral position of the other peaks in MA did not change. The strong spectral intensity peak for all the drug composites for different frequency regions suggests the prepared composite has unchanged crystalline structure. XRD was also carried out for crystal structure determination. X-ray powder diffraction (Figure 7) confirmed that the crystal structure of pure MA and the MA-nGO composites were similar and that there was no change in polymorph.

Thermogravimetric analysis (TGA) was used to measure and monitor the decomposition profiles of the nGO composites as well as the pure drug. Figure 8 shows that all the samples decomposed around similar temperatures. No significant variation of structural decomposition was observed up to 700 °C. The analysis suggests that a very stable thermal composite material was produced and could be stable enough in a wide temperature range. Additionally, the incorporated concentrations of nGO for all the composites were also determined by TGA. The analysis of TGA data was carried out using the first derivative of the decomposition profile. The calculated incorporation of nGO for all the composites were found to be around 1.0% (Figure 8b). The data suggest that maximum nGO incorporation during antisolvent precipitation could not be increased further. Figure 9 shows additional thermal analysis data using a differential scanning calorimetry (DSC). This data shows the material’s melting point. The original melting point for pure MA was ~219–220 °C. The prepared MA-nGO showed very similar melting point, suggesting that there was no alteration of the polymorph.

### 2.5. In Vitro Dissolution Study

Figure 10a is the dissolution profile for MA and its nGO composites. It is evident from the profile that nGO helped enhance the dissolution rate and aqueous solubility significantly. As can be seen from the graph, there was also a trend of increasing nGO concentration to a certain point, which also increased the dissolution rate. Table 2 shows the enhanced rate and reduced time taken to reach 50% dissolution (T_5_**_0_**) and 80% dissolution (T_80_), respectively. Despite using larger amounts of nGO to incorporate, only ~1% was embedded. The dissolution profiles of the composites were similar but showed improvement in dissolution behavior compared to pure drug. With the incorporation of 1.04% of nGO, T_50_ and T_80_ went down from 138.0 min to 27.0 min and did not dissolve for 97.0 min, respectively. Pure MA active ingredients have very low solubility, so with the incorporation of nGO, it was possible to dissolve 100.0% of the ingredients. Similarly, the initial dissolution rate (0 min to 20 min) increased with nGO incorporation; from 14.41 µg/min for pure drug to 60.62 µg/min when the nGO incorporation concentration was 1.04%. The in vitro dissolution study was conducted in a media (0.1 N HCl) that mimicked the gastric pH (pH 1.4). An in vitro study was also carried out in simulated intestinal environment (pH 6.5). The composites showed lower percentages of dissolution and precipitate formation (Figure 10b). Table 3 shows that at the intestine pH, the 50% dissolution average time was around 94 min for the composites, whereas 80% dissolution could not be achieved for any of the composites. The pure MA did not even show 50% dissolution. Additionally, comparable initial (0 min to 20 min) dissolution rate was two- to three-times higher for MA-nGO composites than pure MA. This implies that the MA-nGO composites could potentially dissolve in vivo and enhance absorption.

## 3. Materials and Methods

### 3.1. Materials

Megestrol acetate was obtained from commercial vendor TCI America (lot. JEAEN-EM). Dimethyl Sulfoxide (DMSO), 1-Octanol, and methocel E4M (HPMC) were bought from Sigma Aldrich, St. Louis, MO, USA. Graphene oxide (4.0 mg/mL) liquid water dispersion (~10 µm particle size, 0.05 wt.% monolayer content, purity > 95%) was purchased from Graphenea (Graphenea Inc., Cambridge, MA, USA). Water used in the experiment was purified with Milli-Q plus system.

### 3.2. Synthesis of Nanographene Oxide (nGO) Suspension

Different volumes of graphene oxide (4.0 mg/mL) water dispersion (0.65 mL, 1.0 mL and 1.5 mL) were diluted by adding 20 mL Milli-Q water and sonicated for 1.0 h using a high-power probe sonicator. The solution was monitored with a temperature sensor and maintained at 40 °C. The power was set to 70% of 800 W capacity. After sonication, the solution was brought to room temperature and immediately analyzed on a Zetasizer dynamic light scattering (DLS) instrument for controlled particle size analysis. The particle size was measured before and after sonication.

### 3.3. Incorporation of (nGO) into MA via Antisolvent Precipitation

An antisolvent technique was used for drug composite synthesis based on a published procedure [52]. Different amounts of nanographene oxide (2.6 mg, 4.0 mg, and 6.0 mg) were used during the incorporation process to prepare the MA sample solution. DMSO was used as the organic solvent to dissolve the MA. A liquid suspension of 0.87% (0.65 mL), 1.33% (1.0 mL), and 1.96% (1.5 mL) of nGO was added dropwise to the drug solution (300.0 mg drug) in the ratio of 1:115, 1:75, and 1:50 (nGO: drug) and sonicated for 10 min. Its appearance was clear. The resulting nGO-incorporated (determined by thermogravimetric analysis (TGA)) drug mixture was kept in a fume hood to reach room temperature. Next, the antisolvent agent, Milli-Q water, was added dropwise into the clear solution to precipitate the MA-nGO composites, which formed a dark and cloudy crystalline precipitate (called antisolvent precipitation/crystallization process). The prepared composites were filtered through a 5.0 µm membrane filter for several hours. After filtration, the sample was washed with water several times to remove organic solvent and occasionally checked with litmus paper until neutral pH was obtained. The precipitated sample was then dried in a vacuum oven with 30.0 mm-Hg pressure and 120.0 °C for 48.0 h to reach the constant dried weight.

### 3.4. Physical and Chemical Characterization of Prepared Drug Composites

Several physical properties of the composite were tested for water solubility. We used octanol-water partitioning to test for lipophilicity and DSC for drug melting point and purity determination. Other physical parameters, including particle size of the drugs, nanoparticle, and nanoparticle-drugs, as well as zeta potential for the stability of the drug colloidal suspension, were measured using a Malvern Nano ZS (Worcestershire, United Kingdom) Dynamic Light Scattering Technique. Drug crystal surface morphology was analyzed using JSM-7900F, JOEL (Tokyo, Japan), Scanning Electron Microscope (SEM), where the specimens were carbon coated prior to imaging and elemental composition measured through SEM-EDS analysis. Additionally, chemical surface morphology and inner structure was observed through JEM-F200, Transmission Electron Microscope (TEM) (JOEL, Tokyo, Japan). Alternatively, particle thickness was characterized previously using Bruker Dimension icon Scan Asyst, AFM instrument (Billerica, MA, USA) [52]. The crystalline and chemical functional structure of the prepared MA-nGO composites were analyzed through Raman spectroscopy, which was carried out using a ThermoFisher Scientific DXR Raman Microscope (Madison, WI, USA) with 532 nm wavelength laser and filter. Similarly, precise crystalline structure before (pure MA) and after nGO incorporation (MA-nGO composites) was monitored with powder X-ray diffraction that was performed using PANalytical EMPYREAN XRD (Malvern Panalytical, Malvern, United Kingdom) with Cu Kα radiation source under scanning conditions of 5–70 degrees angular range. Incorporation properties and sample decomposition of the prepared materials were analyzed with TGA with PerkinElmer 8000 (PerkinElmer, Waltham, MA, USA). The thermogravimetric analysis samples were heated from 30 °C to 700 °C under a 20 mL/min airflow at 10 °C/minute heating rate. Sample purity and melting temperature were also monitored using PerkinElmer DSC 6000 (PerkinElmer, Waltham, MA, USA) instrument with a vented sample holder. Finally, in vitro dissolution performance was analyzed through United States Pharmacopeia (USP-42, Distek Inc., North Brunswick, NJ, USA) dissolution system that was also combined with the Agilent 8453 UV-Vis (Santa Clara, CA, USA) instrument for the observation of drug concentration changes at 2 different pH media.

### 3.5. In Vitro Dissolution Testing Methods

In order to simulate gastric conditions, 0.1 N hydrochloric acid (pH = 1.4) was prepared and used as the dissolution media. Samples were added to vessel with 900 mL of 0.1 N HCl and equilibrated to 37 °C ± 0.5 °C. A paddles technique was operated at 75 revolutions per minute to mix the media at a constant rate. Pre-weighted amounts (40 mg) of powder sample were added to the dissolution media. Approximately 3 mL of sample aliquots was withdrawn at specific times of 5 min, 10 min, 20 min, 30 min, 50 min, 80 min, 120 min, 150 min, 180 min, 240 min, and 300 min using a syringe. The aliquot was then filtered with a 0.2 µm syringe filter and analyzed in UV-Vis instrument to determine the concentration of dissolved MA composite at a wavelength of 298 nm. The analysis was averaged from the triplet measurement of sample composites.

A second media was prepared which consisted of 0.01% sodium dodecyl sulfate or SDS solution (0.1 mg/mL) at a pH 6.5. This was to investigate intestinal environment and establish in vivo efficacy. Similar paddle method was applied with non-sink conditions (dissolution media is lower than needed to dissolve drug) for a direct evaluation of precipitation or supersaturation behavior. These conditions provide the most insight in vivo. We tested our composites’ ability to generate and maintain supersaturation by first suspending a high amount of drug (150 mg) in 15 mL of 0.5% methocel E4M (water soluble HPMC polymer as the suspending agent). Drug suspension (10 mg/mL) was then added to dissolution flask filled with 500 mL of simulated intestinal media. Samples were taken out over specified points in time and analyzed by UV-Vis spectroscopy. The experiment was also repeated with 1-week aged suspension.

## 4. Conclusions

The incorporation of nGO significantly enhanced the dissolution rate and solubility of MA. The investigated work for antisolvent precipitation was successful for producing nGO-drug composites. The analyzed SEM and TEM images showed nGO embedded and adhered to the surface of the drug, allowing for interaction with water molecules. The images also showed that the formulated crystal structure was the same size and MA did not appear amorphous after sonication. Raman spectroscopy, XRD, and DSC showed that the presence of nGO did not changes the polymorph or melting point. Increased aqueous solubility and octanol-water partition coefficients were observed for the MA-nGO composites. The increase in dissolution rate was significant with nGO incorporation and the T_50_ and T_80_ values were significantly lowered. The observed data suggest that hydrophilic channels were produced onto drug crystals during nGO incorporation via antisolvent precipitation.

This research provides an alternative route compared to conventional techniques for increasing the solubility of oral hydrophobic drugs. MA, previously improved through solid dispersion formulations, was successfully incorporated with nGO to enhance dissolution rate. This work shows that it is possible to avoid expensive approaches that alter crystal structure and suffer from low drug load. The technique requires minimal amount of nanomaterial and small volumes of solvent, has high yield, and is scalable. These initial results are promising, and further work will be conducted to apply this technique to other hydrophobic compounds with low dissolution rate.

## Figures and Tables

**Figure 1 molecules-26-01972-f001:**
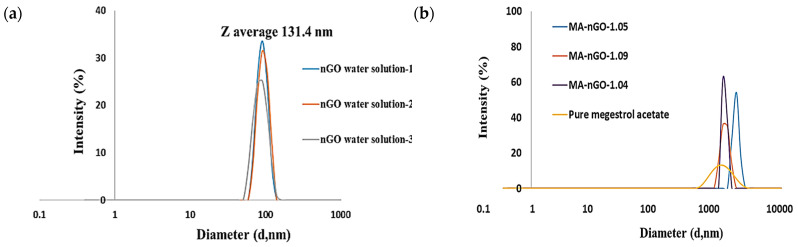
Zetasizer particle size measurement of (**a**) nano graphene oxide (nGO) and (**b**) megestrol acetate (MA)-nGO composite.

**Figure 2 molecules-26-01972-f002:**
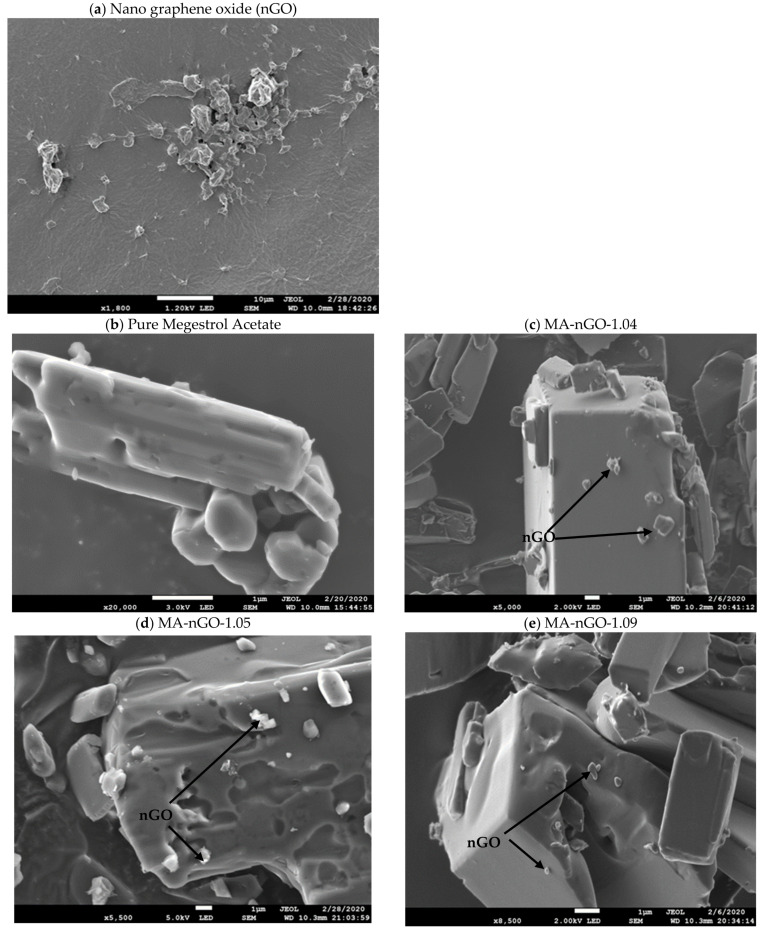
Scanning Electron Microscope (SEM) images of (**a**) nano graphene oxide, (**b**) pure megestrol acetate, (**c**) MA-nGO-1.04, (**d**) MA-nGO-1.05, and (**e**) MA-nGO-1.09.

**Figure 3 molecules-26-01972-f003:**
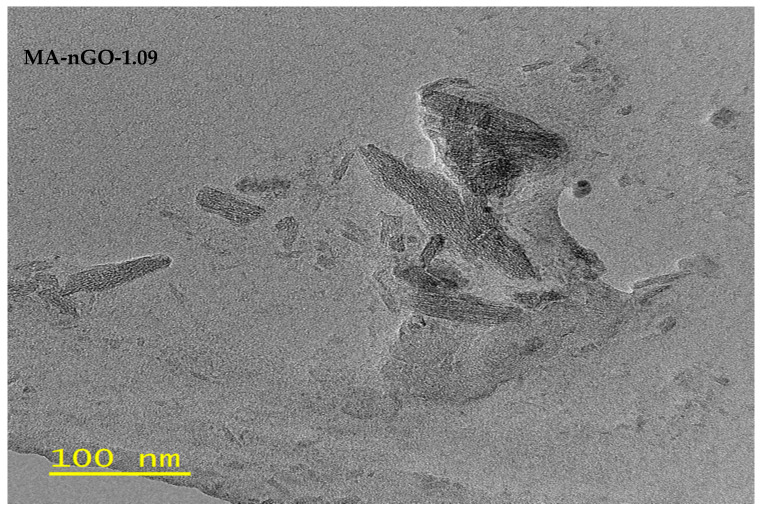
Transmission Electron Microscope (TEM) image of MA-nGO-1.09.

**Figure 4 molecules-26-01972-f004:**
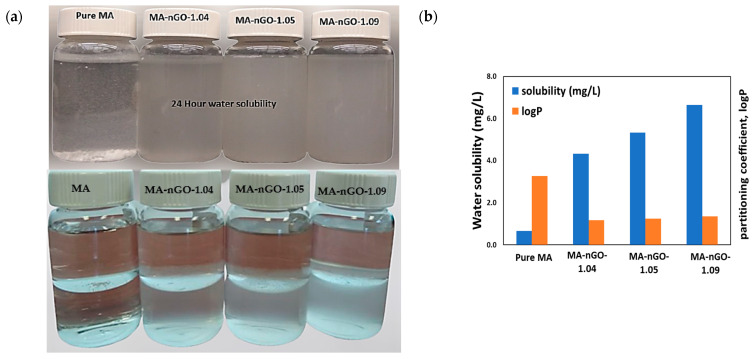
(**a**) Photograph of MA-nGO composites 24-hour solubility in water (upper) and octanol-water partitioning (lower). (**b**) Water solubility and octanol-water partitioning coefficient (logP) of MA-nGO composites.

**Figure 5 molecules-26-01972-f005:**
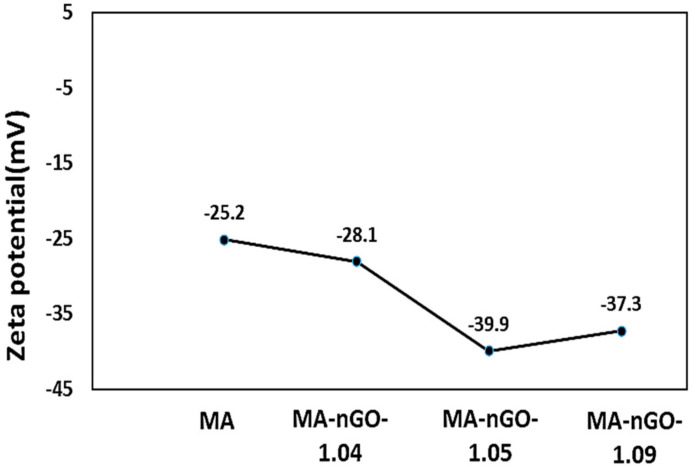
MA-nGO composite particles electrical zeta potential analysis in the Malvern Zetasizer.

**Figure 6 molecules-26-01972-f006:**
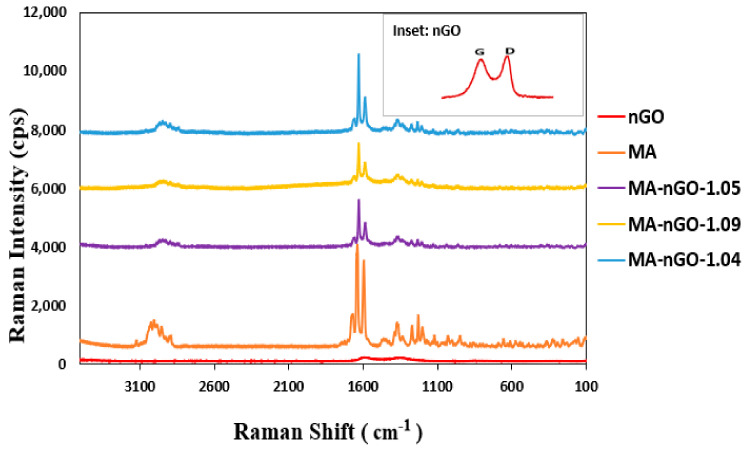
Raman spectra of pure MA and MA-nGO composites.

**Figure 7 molecules-26-01972-f007:**
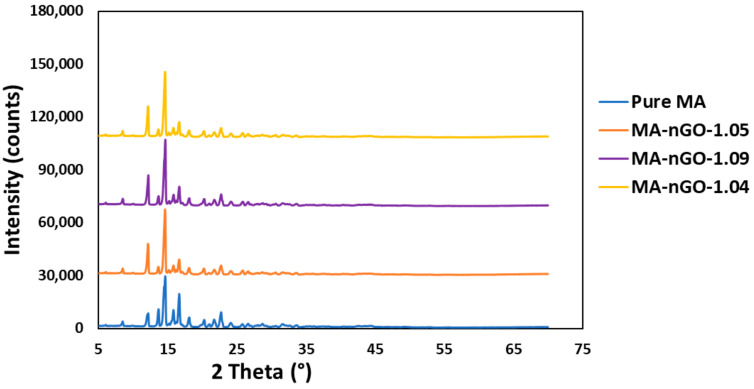
XRD analysis of MA and MA-nGO composites particles.

**Figure 8 molecules-26-01972-f008:**
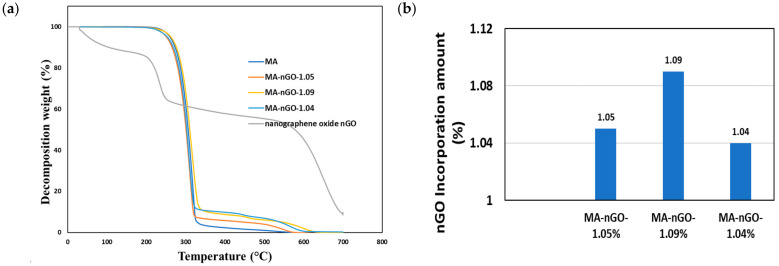
(**a**) Thermogravimetric analysis (TGA) of drug composites and (**b**) computed nGO incorporation.

**Figure 9 molecules-26-01972-f009:**
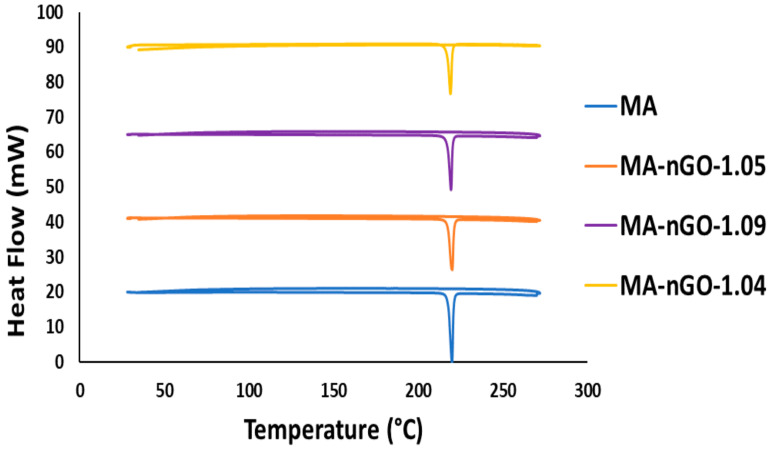
Differential Scanning Calorimetry analysis for MA drug composites.

**Figure 10 molecules-26-01972-f010:**
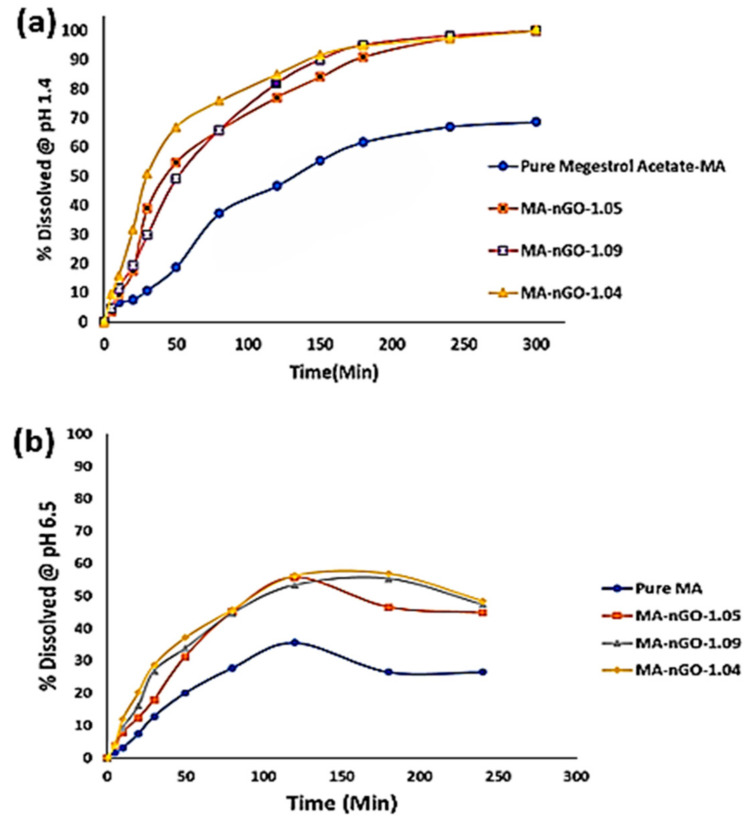
Dissolution profile for MA-nGO-drug composites (**a**) at pH 1.4 and (**b**) at pH 6.5.

**Table 1 molecules-26-01972-t001:** Data table for nano graphene oxide elemental analysis.

Nano Graphene Oxide EDS (Energy Dispersive X-ray Spectroscopy) Quantitative Results	
Carbon (Wt. %)	**52.58**
Oxygen (Wt. %)	**47.42**

**Table 2 molecules-26-01972-t002:** Dissolution data at gastro pH (1.4) conditions.

Drug	50% Dissolution Time (T_50_)	80% Dissolution Time (T_80_)	Initial Dissolution Rate (µg/min)	Melting Points (°C)
MA	138	** (not dissolved)	14.41	219.8
MA-nGO-1.05	42	130	28.54	219.1
MA-nGO-1.09	49	112.5	33.05	219.2
MA-nGO-1.04	27	97	60.62	218.8

**Table 3 molecules-26-01972-t003:** Dissolution data at intestine pH (6.5) conditions.

Drug	50% Dissolution Time (T_50_)	80% Dissolution Time (T_80_)	Initial Dissolution Rate (µg/min)
MA	** (not dissolved)	** (not dissolved)	84.1
MA-nGO-1.05	93	** (not dissolved)	128.2
MA-nGO-1.09	97	** (not dissolved)	154.6
MA-nGO-1.04	92	** (not dissolved)	232.7

## Data Availability

Data will be available from the authors upon request.

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
