# Peer review of "Hydrophilic and Functionalized Nanographene Oxide Incorporated Faster Dissolving Megestrol Acetate"

_molecules, 2021, doi:10.3390/molecules26071972_

Round 1

Reviewer 1 Report

The main critical issue of the work were resolved by the authors by performing additional experiments in media simulating the intestinal fluids, thus the manuscript can be published in the present form.

Reviewer 2 Report

Today I have got acquainted with the revised manuscript and the authors 
improved it a lot. They added new data and revised the unclear issues.  Improvement effort was considerable and the manuscript could be considered for publication.

This manuscript is a resubmission of an earlier submission. The following is a list of the peer review reports and author responses from that submission.

Round 1

Reviewer 1 Report

In this work, nGO-MA complex was prepared to enhance the drug solubility in aqueou medium.

the manuscript needs some improvements, as listed below:

1) Line 72: endothermal …cancer treatment: correct with endometrial?

2) Line 77: correct formulation with “formulations”.

3) Line 102: “Sample composites were prepared using different concentrations (%) of nanographene oxide for incorporation”: the synthetic procedure has to be improved specifying the amount of nGO and the nGO/drug ratio.

4) The authors should explain why the dissolution tests were performed only at acidic pH. For an oral formulation, indeed, the increased solubility in aqueous media should be evaluated also in media mimicking the pH of intestinal environment.

Reviewer 2 Report

The manuscript reports results from a study on preparation of composites from the drug  megestrol acetate (MA) and nanographene oxide (GO) aimed at increasing the solubility of the bioactive component, i.e. a result that might have practical impact. However, the study does not carry high level of originality. There has already been published a similar investigation (see ref. 52) concerning other bioactive compounds but applying the same approach of co-precipitation with GO.

Moreover, there are unclear or general explanations of the observations in addition to a large number of technical errors or not refined phrasing that decrease the quality of the manuscript. I would like to give several examples to illustrate the above said without covering all issues that have to be addressed:

  1. There are abbreviations in the abstract without being introduced
  2. The authors described the formation of the composite precipitate applying antisolvent technique as incorporation of nGO (without specifying where, see lines 11, 16, 17 and further) or as incorporation into MA (line 100). Both explanations are not precise.
  3. In line 20 it is said that the functionalized nanoparticles are incorporated in the drug crystal lattice which is the cause for the increased solubility of the bioactive component. Though it seems to bring insight in the obtained results, this is also not proved by the experimental data. The authors conclude that the MA crystal structure remained unchanged based on the XRD, Raman spectroscopic and DSC data.
  4. The authors wrote that they “have reported the development of drug-GO composites where the GO was effective in enhancing the dissolution of antifungal agent griseofulvin and anti-inflammatory agent Ibuprofen [35,39,40]” but the cited publications are co-authored by other researchers.
  5. The units of the size of the GO nanoparticles in the Figure 1 are (d, nm) and it is not explained what the authors mean under Z average. The left graph does not have a legend and the three measurements (curves in different colours) cannot be assigned to certain samples or experimental conditions.
  6. There are unclear sentences such as “The experimental formulation found variation in structural morphology from their original drugs due … “ (lines 77-78) or “The particle size of GO must be in the nano scale in order to be incorporated without covering the whole crystal” (lines 147-148) that need to be revised.  
  7. The results from the dissolution measurements are presented in Figure 10 and Table 2. The comments say that MA is practically insoluble while the active component is released in 100% from the composite material. In fact the Figure 10 displays that 60 % of the pure drug has been dissolved within the time frame of the experiment.
  8. It is not clear how the concentration of nGO in the compositions was calculated from the TGA curves. The authors report that “the incorporation concentration for all the prepared drug composites found were approximately 1.0 %.” So, if it is correct then next question arises – what is the difference between the three composites analysed and discussed: MA-nGO-0.87, MA-nGO-1.33 and MA-nGO-1.96?
  9. The authors did not obtained data that could provide the physico-chemical basis to explain the observed behaviour of the drug with the exception of the zeta potential data and they do not search for other techniques. Even analysis of the size of MA crystallites prior and after co-precipitation with nGO was not done on the basis of the XRD data.

In conclusion: The manuscript needs a precise and thorough revision, deeper discussion and if possible application of other techniques or at least the obtained experimental data to be re-analyzed.